# TRL: Discriminative Hints for Scalable Reverse Curriculum Learning

## Abstract

Deep reinforcement learning algorithms have proven successful in a variety of domains. However, tasks with sparse rewards remain challenging when the state space is large. Goal-oriented tasks are among the most typical problems in this domain, where a reward can only be received when the final goal is accomplished. In this work, we propose a potential solution to such problems with the introduction of an experience-based tendency reward mechanism, which provides the agent with additional hints based on a discriminative learning on past experiences during an automated reverse curriculum. This mechanism not only provides dense additional learning signals on what states lead to success, but also allows the agent to retain only this tendency reward instead of the whole histories of experience during multi-phase curriculum learning. We extensively study the advantages of our method on the standard sparse reward domains like Maze and Super Mario Bros and show that our method performs more efficiently and robustly than prior approaches in tasks with long time horizons and large state space. In addition, we demonstrate that using an optional keyframe scheme with very small quantity of key states, our approach can solve difficult robot manipulation challenges directly from perception and sparse rewards.

## 1 Introduction

Reinforcement learning (RL) aims to learn the optimal policy of a certain task by maximizing the cumulative reward acquired from the environment. Recently, deep reinforcement learning has enjoyed great successes in many domains with short-term and dense reward feedback (Mnih et al., 2016) (e.g. Atari games, TORCS). However, many real world problems are inherently based on sparse, binary rewards, where the agent needs to travel through a large number of states before a success (or failure) signal can be received. For instance, in the grasping task of a robotic arm, the most accurate task reward would be a binary reward only if the agent successfully picked up the the object or not. This kind of goal-oriented tasks with sparse rewards are considered the most difficult challenges in reinforcement learning. That is, the environment only provides a final success signal when the agent has accomplished the whole task. The search space of these tasks may exponentially expand as state chain extends, which adds to the difficulty of reaching the final goal with conventional reinforcement learning algorithms.

There have been multiple lines of approaches for tackling such sparse reward problems, including the ideas based on intrinsic motivation (Stadie et al., 2015; Bellemare et al., 2016; Houthooft et al., 2016), hierarchical reinforcement learning (Florensa et al., 2017a; Kulkarni et al., 2016), curriculum learning (Karpathy et al., 2012; Sukhbaatar et al., 2017) and experience-based off-policy learning (Andrychowicz et al., 2017). Recently, a particularly promising approach is the idea of reverse curriculum generation (Florensa et al., 2017b). By reversing the traditional reinforcement learning process and gradually expanding the set of start states from the goal to the starting point, this method does achieve substantial progress in some simulation tasks. However, the newly sampled start set mixes old and new states, which leads to inefficiency in training, as the agent spends a large amount of time reviewing rather than learning new skills. This problem aggravates in large state space tasks where the expansion could be really slow, and the storing of history data is also impracticable due to the memory limitation.

To solve goal-oriented tasks and avoid the drawbacks mentioned above, we propose a Tendency Reinforcement Learning (TRL) architecture which shapes a reward function with former experience and use it to stabilize and accelerate training. To achieve this, a discriminative reward (the "tendency") is shaped to output a judgment on the success tendency of each state, and the reward function for the agent is hybrid, which combines the final goal reward with the tendency hints. We define a set of start states as a phase, and continue extending it from the final goal, until it reaches where the task starts. In each phase, the tendency reward influences the agent's training. After training each phase, the tendency reward is updated using collected experience. This mutual promotion contributes to a dramatic improvement in training efficiency and no longer requires keeping all history data.

In this paper, we introduce three novel techniques: First, we propose a hybrid reward design that combines the final reward with tendency hints. Secondly, with the introduction of a phase administrator, the curriculum for each phase is automated, which increases the efficiency. Lastly, we develop an optional keyframe scheme, that is, our framework is also compatible with additional keyframes with no strict accuracy requirement, as long as they are effective at reducing irrelevant search space.

The major contribution of this work is that we present a reliable tendency reinforcement learning method that is capable of training agents to solve large state space tasks with only the final reward, taking raw pixels as perception. In our experiments, we apply and test the model in diverse domains, including a maze game, Super Mario Bros and 3 simulated tasks (a robotic grasping task, a crane conveyance task and a pick-and-place challenge), where it proves more efficient and competitive than the prior works.

## 2  RELATED WORK

Recently, great progress has been made in RL by approximating the policy and value functions through neural networks (Mnih et al., 2013; 2015; Schulman et al., 2015; Mnih et al., 2016). However, these RL methods often suffer from extremely slow training progress or fail to learn in sparse reward tasks. There have been several lines of approaches to tackling such problems, namely curriculum learning, hierarchical RL, intrinsic motivation and demonstration-augmented RL, each with its own constraints and assumptions.

Curriculum learning works by deconstructing a difficult, long-horizon learning task into a sequence of easy to hard tasks (Karpathy et al., 2012). While classic curriculum is often defined manually, recently in RL a number of work has shown promising results automated curriculum generation (Pinto et al., 2017; Sukhbaatar et al., 2017; Wulfe & Hershey, 2017; Florensa et al., 2017b). Our work is closest to this line of work. In particular, reverse curriculum generation (Florensa et al., 2017a) has proposed a very similar curriculum generation through exploration backwards from the goal states, and has shown promising results in solving a variety of sparse reward problems. However, the critical difference of our approach from theirs is that we propose the use of the tendency reward trained discriminatively to summarize the past histories of successful and unsuccessful states, allowing our algorithm not to store the whole history of starting sets and further provide a dense shaping reward. Empirically our design choice is found to scale to larger state tasks with longer planning horizon better than this prior approach.

Hierarchical RL and intrinsic motivations are two other approaches for tackling the sparse reward problems. Hierarchical RL (Sutton et al., 1999; Kulkarni et al., 2016) learns to plan using a hierarchical of low-level policies, often directly provided to the high-level planner, to drastically reduce the search dimension of the problem, while intrinsic motivation often provides additional dense rewards in terms of novelty (Stadie et al., 2015; Houthooft et al., 2016; Bellemare et al., 2016) or some notion of empowerment (Mohamed & Rezende, 2015) to improve the exploration under no rewards. However, both approaches assume forward-based exploration to solve the sparse reward task, which becomes increasingly difficult as the dimensionality and the time horizon increase, unless strong structured prior is provided in HRL. Our method, with the similar assumptions as in reverse curriculum learning, allows much more efficient learning and exploration.

Lastly, there are a number of work that utilizes demonstrations to tackle such problems. Inverse reinforcement learning infers the underlying reward function from expert demonstrations(Ng et al., 2000; Abbeel & Ng, 2004; Ziebart et al., 2008), and recent work has used such reward function as

potential-based shaping reward to accelerate learning (Brys et al., 2015; Harutyunyan et al., 2015). Another line of work uses demonstrations that consist of keyframes to do Keyframe-based Learning (Akgun et al., 2012). These methods, however, make strong assumptions about the quality of demonstrations they receive. By contrast, our optional keyframe scheme only requires a small number of key states instead of several full demonstration trajectories, and is robust to noises. Specifically, for a challenging robotic manipulation task from perception, we only required 5 key states to solve the task.

# 3 PRELIMINARIES AND ASSUMPTIONS

Table 1: List of symbols

| Symbol | Definition |
|---|---|
| $\mathcal{P}_i$ | Set of start states in $i^{th}$ phase. |
| $\mathcal{P}^{task}$ | The task start phase. |
| $K$ | Number of states in each phase. |
| $O$ | Observation acquired from environment. |
| $s$ | State that the agent arrives at. |
| $s^g$ | State where the agent reaches the goal. |
| $r^f$ | Final goal reward(binary). |
| $\pi$ | Policy pursued by the agent. |
| $T(\cdot)$ | Function of tendency reward. |
| $r(\cdot)$ | Hybrid reward function combining $T(\cdot)$ and $r^f$ |
| $\alpha$ | Lower bound of success rate in phase extension. |
| $\beta$ | Upper bound of success rate in phase extension. |
| $N_i$ | Number of training iterations in $\mathcal{P}_i$. |
| $x_i$ | Extension step size after the training of $\mathcal{P}_i$. |
| $x_{max}$ | Max random walk steps in extension. |
| $d$ | Number of history frames in Gaussian kernel. |

**Algorithm 1:** TendencyRL

**input** : Goal state $s^g$, starting phase $\mathcal{P}_1$, task start phase $\mathcal{P}^{task}$.

**output:** Trained policy $\pi$.

Phase index $i \leftarrow 1$;
**if** $\mathcal{P}_1 = \varnothing$ **then**
 $\quad \lfloor\ \mathcal{P}_1 \leftarrow SampleNearby(s^g, K)$;
Train policy $\pi$ beginning from $\mathcal{P}_1$ with environmental reward;
Collect trajectories & train Tendency Reward $T(\cdot)$;
**while** $\mathcal{P}_i \cap \mathcal{P}^{task} = \varnothing$ **do**
$\quad$ $\mathcal{P}_{i+1} \leftarrow$
$\quad$ $PhaseAdministrator(\mathcal{P}_i, N_i)$;
$\quad$ Train policy $\pi$ beginning from $\mathcal{P}_{i+1}$ with hybird Reward;
$\quad$ Update Tendency Reward;
$\quad$ $i \leftarrow i + 1$;
**return** $\pi$;

## 3.1 PRELIMINARIES

We revisit the conventional reinforcement learning setting where an agent interacts with the environment $Env$ sequentially. At each time step $t$, the agent receives an observation $O_t$ and chooses an action $a_t$ from a set of action space $\mathcal{A}$ according to its policy $\pi$. Then the environment will give the agent the next state $s_{t+1}$ and a reward $r_t$. The goal of the agent is to maximize the expected return from each state $s_t$ through accumulating the reward by $R_t = \sum_{k=0}^{\infty} \gamma^k r_{t+k}$.

## 3.2 ASSUMPTIONS

The tendency reinforcement learning model is based on three assumptions that follows (Florensa et al., 2017b) and hold true for a variety of practical learning problems.

**Assumption 1.** *We can arbitrarily reset the agent into any start state $s \in \mathcal{P}_i$.*

**Assumption 2.** *At least one state $s^g$ is provided such that $s^g \in S^g$.*

**Assumption 3.** *The Markov Chain induced by taking uniformly sampled random actions has a communicating class[1] including all start states in $\mathcal{P}^{task}$ and the given goal state $s^g$.*

Assumption 1 can be satisfied in many robotic manipulation problems by storing the start states in $\mathcal{P}_i$ with low-dimensional data (e.g. angle of joints and velocity of motors). Acquiring these data and sending commands to let robotic apply these settings are widely provided by standard robotic APIs, and such functions are similar in game engines. This assumption enables the agent to train from

---

[1]A *communicating class* is a maximal set of states $C$ such that every pair of states in $C$ communicates with each other. Two states *communicate* if there is a non-zero probability of reaching one from the other.

recommended start states, which proves to increase learning efficiency (Kakade & Langford, 2002). Combining Assumption 1 with Assumption 2, we are able to reset the state to $s^g$. Assumption 3 ensures that there exists a trajectory formed from a sequence of actions from any start state in $\mathcal{P}^{task}$ to $s^g$, and vice versa. This assumption can be satisfied by many robotic manipulation problems and games, as long as there are no major irreversibilities in the system.

Symbols used in our paper are listed in table 1 for reference.

## 4 THE MODEL

Naïve applications of conventional RL tend to fail in **goal-oriented tasks** with a large state space and sparse rewards. However, this kind of tasks are common in real world. TRL is able to tackle them by reducing aimless searching with the application of 1) a discriminative tendency reward shaping that utilizes the agent's former experience; 2) a phase administrator that assigns customized curriculum to the agent; 3) a keyframe scheme that shrinks search space and accelerates learning in large state space multistage tasks (optional).

Our key idea is to let the agent utilize previous experiences through a tendency classifier trained on history trajectories in a supervised manner (Sec. 4.1). We define each phase as a set of start states. In training, curriculum are automatically adjusted by a phase administrator (Sec. 4.2). In each phase, the agent learns from rewards produced by the tendency classifier, and trains the classifier with new experiences. The overview of training framework is shown in Fig. 1 and the corresponding algorithm is sketched in Alg. 1.

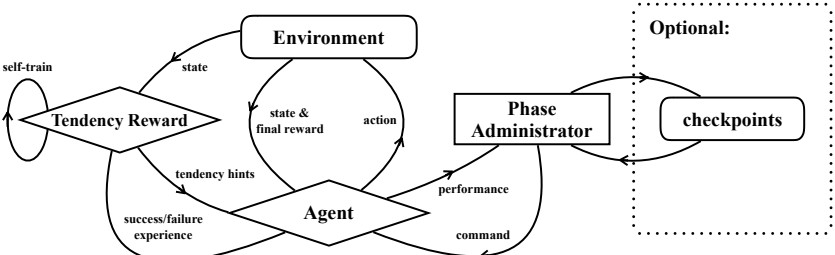

Figure 1: The schematic illustration of TRL.

### 4.1 TENDENCY REWARD

The tendency reward in our model is a binary classifier (e.g. SVM, NN) that takes an observation $O$ as input and outputs a judgment on whether that state leads to the final success. We name it tendency reward mainly because it provides slight guiding hints to speed up the training process in each phase and those large state space tasks thus become trainable.

In training, each time when the agent steps onto a state, it would refer to the tendency reward for a judgment. If the state is evaluated as "favorable", the agent would receive a positive tendency hint as a reward. Otherwise, a negative reward would be issued. To prevent the agent from stalling at a favorable state for unlimited tendency rewards, we defined that such revisited states do not provide any reward, and applied a Gaussian kernel to detect them. The kernel is used with $d$ most recent frames in a bid to detect repeated states, as is shown in Eq. (2):

$$k(x, y) = exp\Big( - \sum_j \min(\max((x_j - y_j)^2 - \delta, 0), 1)/\sigma \Big) \tag{1}$$

$$\rho(s^{now}) = \max_{1 \le i \le d} k(s^{now}, s^i), \tag{2}$$

We assume that $s^{now}$ is a revisited state if $\rho$ is larger than 0.9. The parameter $d$ is 20% of the max steps of each environment and $\sigma = 1$ & $\delta = 0$ are constants in our experiments.

The binary tendency classifier trains on both success and failure trajectories and we use logistic loss as the classifier loss. The hybrid reward is defined in Eq. (3).

$$r(s) = \lambda T(s) + r_f(s), \tag{3}$$

where $\lambda$ is a scale factor ($10^{-3}$ in our experiments) based on the following hypothesis: a positive reward received from the environment should be more important than the output of $T(\cdot)$. Since the tendency hints are very subtle compared to the final reward, the agent's desire to achieve the final goal won't be interfered with.

## 4.2 PHASE ADMINISTRATOR

Each time when the agent has made some progress in $\mathcal{P}_{current}$ (being able to accomplish the task starting from $80\%$ of start set), the $x_{current}$ for the generation of $\mathcal{P}_{current+1}$ will be adjusted by the phase administrator based on the agent's performance.

During the $i^{th}$ phase generation process, the agent start from $\mathcal{P}_i$ and randomly take $x_i$ steps. The newly reached states would then be included into $\mathcal{P}_{i+1}$ if the agent's success rate on that state is between a lower-bound $\alpha$ and an upper-bound $\beta$, which guarantee the new states to a proper difficulty ($\alpha$ is 0.1 and $\beta$ is 0.9 in our experiments).

In each phase generation, a reasonable $x_i$ is required. Large $x_i$ means less phases, while it might increase training iterations in $\mathcal{P}_{i+1}$. Therefore, in order to strike a balance, the phase administrator adjusts $x_i$ according to the number of training iterations $N_i$ in $\mathcal{P}_i$. This adjustment is achieved with a sigmoid function shown in equation (4).

$$x_i = \left\lceil x_{max} \times Sigmoid\left(\bar{N} - N_i\right) \right\rceil = \left\lceil \frac{x_{max}}{1 + e^{N_i - \bar{N}}} \right\rceil, \tag{4}$$

where $\bar{N}$ is the average of $N_1, N_2, \ldots, N_{i-1}$.

The pseudo code of the phase administrator is sketched in Algorithm 2 in Appendix. B.

## 4.3 OPTIONAL KEYFRAME SCHEME FOR EXTREME LARGE STATE SPACE TASK

In tasks with a vast state space, it is usually more difficult for $\mathcal{P}$ to reach any state $s$ nearby $\mathcal{P}^{task}$. Those domains are difficult for both prior approaches and tendency RL. In those cases, we show that by providing just a few *keyframes*, our tendency RL algorithm can take advantage of them to shrink the search space. Ideally, the keyframes should be precise and indicate important states for the task, and in practice fine hand-engineering is often involved. However, for TRL, several rough key states would be sufficient. The agent still extends its phases with a regularly, while after each extension, it additionally checks whether the newly generated phase has covered some states nearby a certain key state (by a Gaussian kernel approximation function as is described in Eq. (5)). If so, the next phase would be directly sampled from that key state rather than generated form $\mathcal{P}_{current}$. In this way, the search space can decrease remarkably.

$$J(\mathcal{P}_{current}) = \sum_{s \in \mathcal{P}_{current}} \max_{t \in Sample(s_{k_i}, K)} k(s, t) \tag{5}$$

If $J$ is larger than 30% of $K$, $\mathcal{P}_{current}$ is believed to cover some states around $s_{k_i}$. Then the next phase is sampled nearby $s_{k_i}$ rather than extended from $\mathcal{P}_{current}$. Pseudo code is shown in Alg. 3 in Appendix. B.

Note that our principle is distinct from LfD (*learn from demonstration*) methods in that the agent is not eager to follow the keyframes and doesn't get additional reward from them. In fact, these small number of keyframes are only used to shrink the search space, and our method also shows robustness to imperfect keyframe settings in experiment where we deliberately mix misleading states into the keyframes, and found that the influence is not significant (Sec. 5.3).

## 5 EXPERIMENT

To evaluate our model, we design five experiments[2] with different levels of difficulties (Fig. 2). In all of these experiments, we apply A3C (Mnih et al., 2016) as the RL solver. The details of environments and network are elaborated in Appendix. C and D.

---

[2]Demo videos of our experiments: https://sites.google.com/view/tendencyrl

| (a) Maze | (b) Super Mario | (c) Grasping Task | (d) Conveyance | (e) Pick-and-place |

Figure 2: Five environments on which we evaluate our model

We first test the efficiency and validity of our model. After testifying the correctness of the overall model, we visualize the distribution of tendency hints and the starting sets of phases. Then we focus on the optional keyframe scheme, verify its compatibility to our method and prove our method's robustness to keyframes of different qualities. Finally, we apply our model to long-term manipulation challenges with large state spaces to show our advantages.

## 5.1 VALIDITY AND EFFICIENCY OF TRL

The training of the $40 \times 40$ maze task takes around 35000 training steps (1 hour and 40 minutes) with 6 threads on GPU. Then the agent can reach the exit from the start point of the maze with a $9 \times 9$ observation around it. To investigate the influence of tendency reward on training, we conduct an experiment comparing the training effect under three conditions: using tendency reward without history phases, using history phases without tendency reward and using neither tendency reward nor history phases. This is not only an ablation experiment but also a comparison with prior work, since the second condition is basically the reverse curriculum generation learning method (Florensa et al., 2017b). The result shows that our model performs well (Fig. 3). We also show the improvement when using phase administrator which let the curriculum automated (Fig. 4).

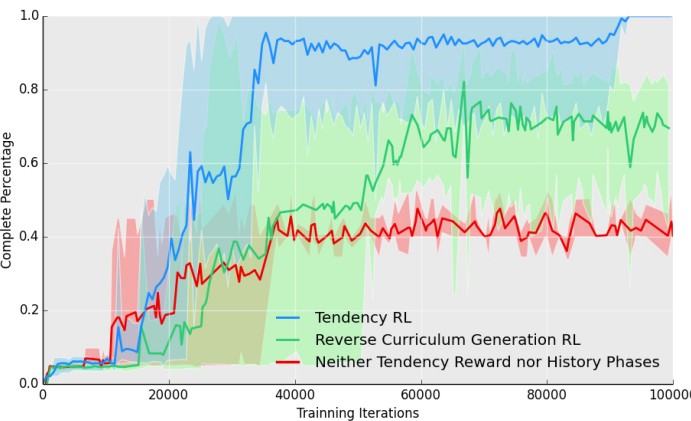

Figure 3: The training results of ablation experiments under three different conditions (mean to standard deviation with 5 random seeds).

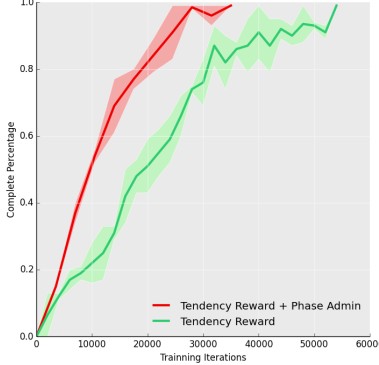

Figure 4: The training results of $40 \times 40$ Maze task with $9 \times 9$ observations. Phase administrator proves to improve training efficiency.

## 5.2 INSPECTING THE LEARNING RESULTS

The agent takes nearly 40000 training steps (around 21 hours) with 8 threads on GPU to learn how to complete the level 1-1 of Super Mario Bros. The policy learned by the agent is shown in Appendix. E.1.2. In order to find out the influence of the tendency reward, we traverse the whole states of some parts of the game and show both of the positive and negative tendency hints (Fig. 5).

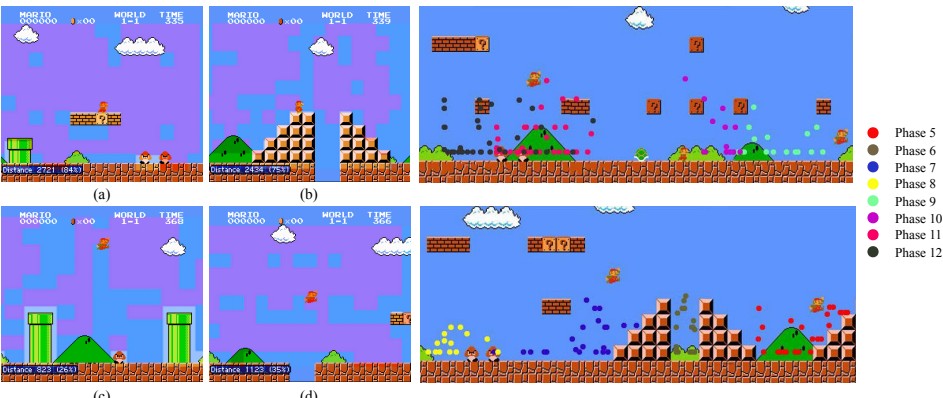

Figure 5: The distribution of tendency hints and generated phases in parts of *Super Mario Bros* game. The purple background color represents positive hints while the blue ones indicate negative hints. The points in the figure indicate the starting points of Mario. Points with same color belong to a same phase.

As we can see in these figures, our agent has acquired the ability to find enemies nearby and dodge or even kill them by jumping up to their heads. Besides, it also recognizes the danger of the pits and can choose the right time to jump over them. These figures show the guiding effect of the tendency reward, which does a great help for the agent when it is far away from the final goal. We also visualize parts of the phases in Mario, which effectively lead the agent to constantly learn new states farther from goal state (Fig. 5).

### 5.3 COMPATIBILITY AND ROBUSTNESS TO OPTIONAL KEYFRAMES

We then enlarge the Maze to $100 \times 100$ also with a $9 \times 9$ observation. With 8 key states as the keyframes, the training takes around 50000 training steps (7 hours and 14 minutes). We also test our agent's robustness to imperfect keyframes, related results can be referred in Appendix. E.3.

We also compare our model with two LfD methods: *data-efficient RL* (Popov et al., 2017) and *potential-based reward shaping (PBRS) from demonstration* (Brys et al., 2015). The former resets the agent randomly to someplace on an expert trajectory while the latter uses an artificial potential reward shaped based on demonstrations. We test them with both *reasonable* and *unreasonable* keyframes / demonstrations. As is shown in Fig. 6, *data-efficient RL* is quite sensitive to the demonstration data and fluctuates constantly facing unreasonable demonstrations. *PBRS from demonstration* can achieve similar performance to *TRL*, but requires a lot more human efforts of hand-engineering its reward function (otherwise it just wouldn't work). Different from these methods, TRL does not need expert demonstration (10 labeled keyframes are sufficient) and requires no further human elaboration. Also, the training of *TRL* is the most stable of three all.

### 5.4 SUCCESS IN LARGE STATE SPACE PROBLEMS

The following experiments mainly focus on illustrating the practicality of our model on solving difficult, real-world robotic manipulation problems with minimal assumptions. When confronted with long-term challenges with large state space, to avoid wasting time on unnecessary states, our keyframe scheme is applied to shrink the search space. In the three experiments, we add less than 15 keyframes to assist training (5 in robotic arm, 5 in conveyance challenge of the crane and 14 in pick-and-place task), and the results are quite satisfying.

#### 5.4.1 GRASPING TASK

The grasping task on robotic arm simulation is a typical large state space problem, where the gripper can move anywhere within the range of the arm. In our design, a camera is fastened to the forepart of the gripper and provides a $64 \times 64$ depth image as the agent's observation. We intuitively design

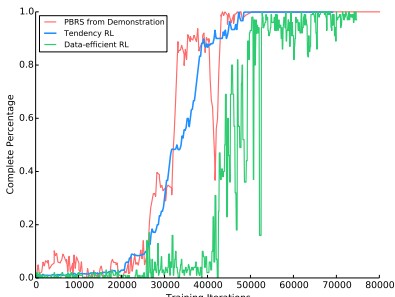 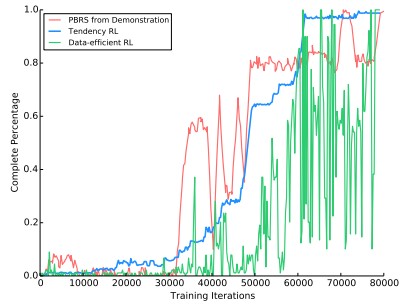

(a) Reasonable keyframes/demonstrations  (b) Unreasonable keyframes/demonstrations

Figure 6: Learning curves of our model, the data-efficient RL and PBRS from demonstration with keyframes / demonstrations of different qualities. Tendency RL is able to achieve a quite satisfactory performance without much human labor.

5 key states and train the whole task for 33360 iterations (4 hours 19 minutes) with 12 threads on GPU, and it turns out that our model can solve the task reliably.

To further investigate the relationship between training efficiency and scale of keyframes, we conduct the same experiment with 5, 8, 11, 13, 15, 18 key points respectively, as is shown in Appendix. E.4.

### 5.4.2   CONVEYANCE CHALLENGE

The conveyance challenge is also a 3D manipulation task with a larger state space. The agent should pick up the ball, take it to correct basket, and drop it right into that basket to receive a positive reward. The camera is also fixed on the gripper that produces $64 \times 64$ RGB images as input. Our model is trained for 99240 iterations (5 hours 2 minutes) with 12 threads on GPU.

Since this task involves multiple operations and has a large state space, we roughly add five key states to help the agent learn different skills, as is shown in Appendix. E.5.

With the intention of finding out the distribution of "favorable" and "adverse" states given by the tendency reward, we make the agent travel through the whole environment at an appropriate height under two conditions: 1) the gripper jaw is open with the target ball fixed on the center of plane; 2) the ball is grasped by the gripper and moves with it. Then we record the tendency reward at each position (Fig. 7).

As expected, areas close to the target basket are assigned the highest tendency hints, which is quite natural as these states are reached before of after a successful drop. On the other hand, locations above those incorrect baskets tend to incur negative tendency hints. More details can be found in Appendix. E.5.

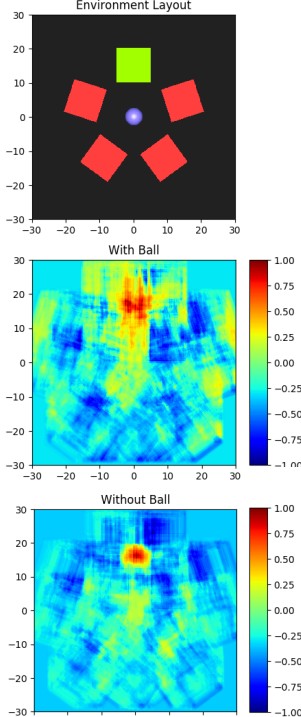

Figure 7: The tendency heat map at the height of 4cm, with different gripper poses in the conveyance task of a crane.

### 5.4.3   RECURRENT PICK-AND-PLACE CHALLENGE

Of all the three experiments in robotic manipulation, this challenge has the largest state space due to endless combinations of the agent's location and orientation and enormous possible positions of plates. This time, the agent observes through a panorama camera that provides $160 \times 32$ first-person point of view.

The whole task can be divided into several identical pick-and-place operations that resemble the conveyance challenge. We provide 14 intuitively designed keyframes to accelerate training (Appendix. E.6). However, these keyframes are not quite sufficient in such a huge state space, and the agent shows confusion from time to time.

Our phase administrator plays a significant role in phase arrangement, and actually makes up for the lack of expert data. Since the agent performs well within each pick-and-place operation, new phases are more concentrated on the intervals between operations.

## 6 DISCUSSION AND FUTURE WORK

We develop a tendency reinforcement learning algorithm that resolves complicated goal-oriented tasks, and evaluate it with multiple experiments based on raw perceptions. Our method consists of two core components: a binary tendency classifier that provides dense hints on whether a state leads to success and a phase administrator that generates a reverse curriculum. In five distinct environments, we find that tendency RL method is efficient and stable enough to converge on a reliable policy within hours even facing intractable problems, and does not require history data which is impossible to acquire in large space domains.

One limitation of our current approach is that our model requires the environment to be capable of resetting to some arbitrary states, which is also a limitation of the reverse curriculum generation method (Florensa et al., 2017b). A promising future direction is to additionally train reset controllers that automatically reset the system to the initial state following each episode, together with a corresponding forward controller (Han et al., 2015).

In future work, our goal is to apply the tendency RL method to real world robotic manipulation tasks with perceptions. Furthermore, we plan to extend the binary tendency classifier into a predictor that outputs the expected success rate on current state, so that our model can also be applied to stochastic environments. Additionally, we intend to release a subset of our code and the trained models to facilitate further research and comparisons.

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

# A  DEFINITIONS AND FORMULAE

## A.1  TERM DEFINITIONS

- **Final goal**: States that indicate the task is successfully finished (e.g. the exit in maze).

- **Phase**: A set of start states at a certain time period.

- **Start state**: A state where the agent starts for the final goal in a certain phase.

- **Success state**: The states leading to the final goal by current policy.

- **Task start phase**: States where the entire task begins, which is the final aim of phases extensions (e.g. the start point in maze).

- **Keyframe**: A small number of key states artificially designed to shrink the search space in large space tasks.

# B  OTHER ALGORITHMS

## B.1  PHASE ADMINISTRATOR

---
**Algorithm 2:** Phase Administrator

---
**input** : Current phase $\mathcal{P}_i$, number of training iterations in the $i^{th}$ phase $N_i$.
**output:** The next phase $\mathcal{P}_{i+1}$.

Calculate $x_i$ from $N_i$ with Equation (4);
$\mathcal{P}_{i+1} \leftarrow \varnothing$;
**while** $|\mathcal{P}_{i+1}| < K$ **do**
    $s_{old} \leftarrow Sample\,(\mathcal{P}_i)$;
    $s_{new} \leftarrow RandomWalk\,(s_{old}, x_i)$;
    **if** $\alpha < SuccessRate\,(s_{new}) < \beta$ *and* $s_{new} \notin \mathcal{P}_{i+1}$ **then**
        $\mathcal{P}_{i+1} \leftarrow \mathcal{P}_{i+1} \cup \{s_{new}\}$;

---

## B.2 TENDENCYRL WITH OPTIONAL KEYFRAMES

---

**Algorithm 3:** TendencyRL with optional keyframes

---

**input** : Goal state $s^g$, key states $s_{k_1}, \ldots, s_{k_n}$, task start phase $\mathcal{P}^{task}$.
**output:** Trained policy $\pi$ for the extreme large space task.

**for** $j$ *from* $1$ *to* $n$ **do**
    $\mathcal{Q}_j \leftarrow SampleNearby(s_{k_j}, K)$;
$\mathcal{R} \leftarrow \bigcup\limits_{j=1}^{n} \{\mathcal{Q}_j\}$;
**while** $\mathcal{P}_{current} \cap \mathcal{P}^{task} = \varnothing$ **do**
    $\pi \leftarrow TendencyRL \left( s_g, \mathcal{P}_{current}, \bigcup\limits_{\mathcal{Q} \in \mathcal{R}} \mathcal{Q} \right)$ (Alg. 1);
    **for** $\mathcal{Q}$ *in* $\mathcal{R}$ **do**
        **if** $\mathcal{Q} \cap \mathcal{P}_{current} \neq \varnothing$ **then**
            $\mathcal{R} \leftarrow \mathcal{R} \setminus \{\mathcal{Q}\}$;
            $\mathcal{P}_{current} \leftarrow \mathcal{Q}$;
            **break**;

**return** $\pi$;

---

## C ENVIRONMENT DETAILS

### C.1 MAZE

Our first environment is a maze task. The size of the whole maze is $100 \times 100$. There is only one exit (final goal) in the middle of the bottom line ($[100, 50]$) and one starting point at $[0, 50]$. The agent is required to find its own way to move from the starting point to exit with a limit observation covering a $9 \times 9$ area around it. At each time-step, the agent has four actions, namely *walk north*, *walk south*, *walk east* and *walk west*, expressed as (N, S, E, W), and transitions are made deterministically to an adjacent cell, unless there is a block, in which case no movement occurs. The environment only gives $1$ point reward when the agent has accomplished the finial goal. On other conditions, the reward remains $0$.

### C.2 SUPER MARIO BROS

The second environment is level 1-1 of Super Mario Bros (Brockman et al., 2016; koltafrickenfer, 2017; asfdfdfd, 2012). The agent should complete the level from start point via a $16 \times 13$ observation, which indicates the nearby situation. When the agent reaches its destination, it would receive $1$ point reward, otherwise it receives $0$ point. The agent has 5 actions: *jump*, *right*, *left*, *jump to right* and *jump to left*, expressed as (J, R, L, JR, JL). Mario could not achieve any reward by killing enemies or collecting coins.

### C.3 GRASPING TASK

The third environment is a grasping task on a robotic arm simulation in MuJoCo. The end effector of the 6-DOF robotic arm is a two-finger gripper with a camera attached between the fingers, which produces $64 \times 64$ overlooking depth images as the agent's observations. In order to manipulate in a more intuitive fashion, poses of joints have been mapped into 3D coordinates, enabling the gripper to move in 6 different directions. There is also a special grasping action that closes the gripper jaws, by which the target cube can be grasped. The agent is initialized randomly above a plane, and $1$ point reward would be received if and only if the target cube placed on the plane is captured by the gripper.

### C.4 Conveyance Challenge

The fourth task is a conveyance challenge of a crane simulated by the physics engine MuJoCo. The agent can interact with a ball placed on a plane using a 3-finger gripper connected to the crane, with 8 actions available (6 for free movements and 2 for gripper manipulations). Surrounding the ball from a distance are some target baskets, among which one basket is randomly selected and colored green. The agent is able to observe through a camera fixed on the gripper that produces $64 \times 64$ RGB images, and only by dropping the ball into the green basket would the agent receive 1 point reward.

### C.5 Recurrent Pick-and-place

The fifth task involves pick-and-place operations simulated in MuJoCo, which requires the agent to stack some plates into a neat pile. This time, the agent observes through a panorama camera that provides $160 \times 32$ first-person point of view. Fixed before the camera is a electromagnet capable of attracting one plate at a time, which is used for plate transfer. Available actions include *turning left / right*, *stepping forward / backward*, *charging / releasing the electromagnet*. The final goal is to gather and pile up all the plates into a straight stack, which is the only approach for the agent to obtain 1 point reward.

## D Network Architectures

### D.1 Maze

The convolution neural network of the policy network has a first layer with 12 $4 \times 4$ filters of stride 1, followed by a layer with 24 $3 \times 3$ filters of stride 1, followed by a layer with 48 $2 \times 2$ filters of stride 1. Then we add a LSTM layer with 384 units which followed by a fully connected layer with 96 hidden units.

### D.2 Super Mario Bros

The convolution neural network of the policy network has a first layer with 48 $4 \times 4$ filters of stride 1, followed by a layer with 48 $3 \times 3$ filters of stride 1, followed by a layer with 96 $2 \times 2$ filters of stride 1. Then we add a LSTM layer with 960 units which followed by a fully connected layer with 240 hidden units.

### D.3 Experiments with MUJOCO Engine

The convolution neural network of the policy network has a first layer with 12 $4 \times 4$ filters of stride 1, followed by a layer with 24 $3 \times 3$ filters of stride 1, followed by a layer with 48 $2 \times 2$ filters of stride 1, followed by a layer with 48 $2 \times 2$ filters of stride 1. Then we add a LSTM layer with 1536 units which followed by a fully connected layer with 96 hidden units. Especially, in the pick-and-place challenge, there is an additional convolution layer with 96 $2 \times 2$ filters of stride 1, and the LSTM layer contains 480 units.

## E Supplementary materials for experiments

### E.1 Super Mario Bros details

#### E.1.1 Training Result

The training result of Super Mario Bros is shown in Fig. 8.

#### E.1.2 Comparison with curiosity-driven method

In Super Mario Bros, we compare our model with another one trained based on curiosity-driven (Pathak et al., 2017). We let the agents randomly move several steps at the beginning before run-

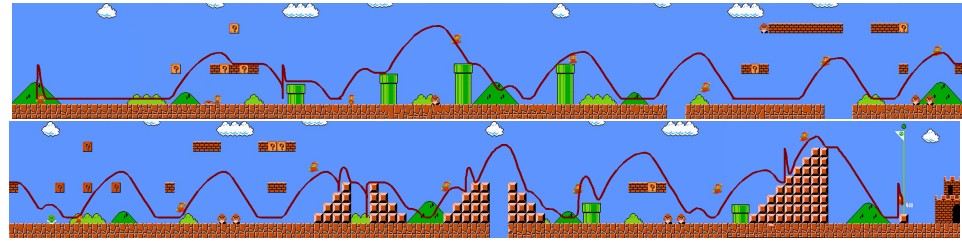

Figure 8: The training result of the Super Mario Bros level 1-1. The red curve indicates the Mario's policy.

ning the models, then compare the completion degree [3] of two models. With the environment only providing a reward when Mario reaches the final goal, we find that our method obtains absolute advantage (Table 2). We also use the model trained in level 1-1 to run the level 1-2 and 1-3 of the game. Since the latter two levels are not trained and require new skills that the first level does not include (Fig. 9), both of the methods may suffer more failures.

Table 2: Compare with the model trained based on curiosity-driven

| Model | Completion degree in level 1-1(%) | Completion degree in level 1-2(%) | Completion degree in level 1-3(%) |
|---|---|---|---|
| ours | 100 | 22 | 17 |
| curiosity-driven | 28 | 1 | 12 |

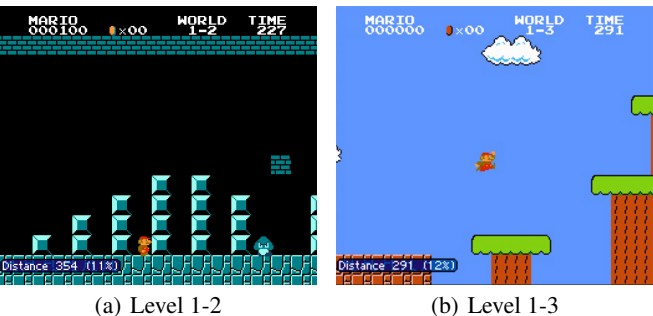

(a) Level 1-2          (b) Level 1-3

Figure 9: The cases where the agent needs new skills in level 1-2 and level 1-3.

### E.2 COMPARISON WITH MAXIMUM ENTROPY IRL

We try the maximum entropy IRL (Ziebart et al., 2008) in the $100 \times 100$ maze task with more than two days' training and about 44G virtual memory used, but it still failed to form a feasible reward function. Since there are 10,000 states in the maze, the traditional IRL methods seem to suffer from the bottleneck of computing ability.

### E.3 ROBUSTNESS TO IMPERFECT KEYFRAMES

To test our model's robustness to imperfect keyframe settings, instead of arranging the keyframes onto the optimal trajectory, we set some of them onto a byway. We choose 8 special phases sampled

---

[3]The mean length they move divided by all length of current level

from the misleading keyframes and show the policy the agent has learned in these phases. As is shown in Fig. 10, in the sixth phase, the agent refuses to detour from the left side and finds a shortcut direct to the exit. A possible explanation is that our tendency reward only provide slight guiding hints which do not bother the agent's exploration for new policies.

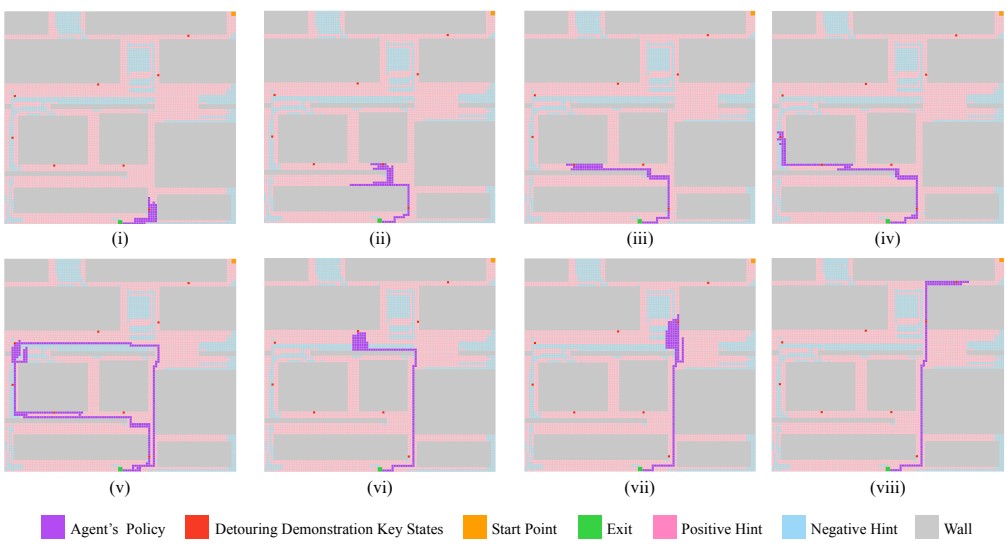

Figure 10: The agent's policies in 8 special phases sampled from misleading key states. The purple points are collected by combining the testing results for 30 times. In the end the detouring keyframes are completely neglected by the agent.

### E.4 THE INFLUENCE OF KEYFRAME SCALE ON TRAINING EFFICIENCY

We conduct the grasping experiment with 5, 8, 11, 13, 15 and 18 keyframes respectively. The influence of keyframe scale on training efficiency is demonstrated in Fig. 11.

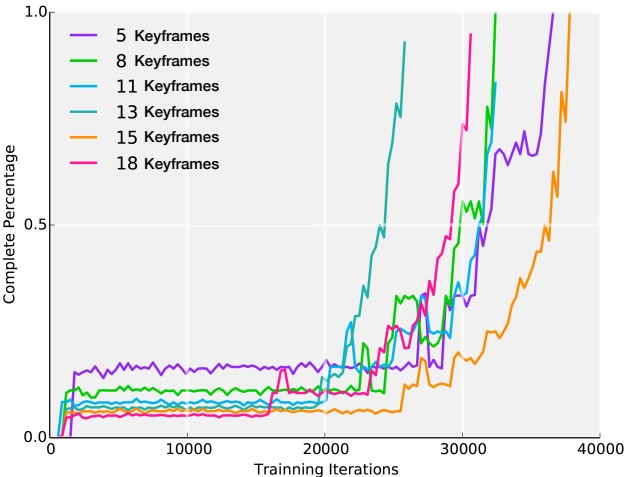

Figure 11: Learning curve of the grasping task with different numbers of keyframes. The training efficiency is relatively higher when 13 keyframes are used. If the number of keyframes is too large or too small, training tends to become less efficient, but still stable.

### E.5 CONVEYANCE CHALLENGE DETAILS

An interesting observation in conveyance challenge is, the areas with positive tendency hints are more concentrated and the boundary between "favorable" and "adverse" states is more explicit when the gripper jaw is open. This is because the states where the agent is not holding the ball are often related to grasping and dropping, which require more accuracy than just transferring the ball.

This experiment also proves that our agent has superb robustness to imperfect keyframe settings. In our previous assumptions, the agent should rise to a proper height where all baskets were completely in view, before moving towards the correct basket (Fig. 12). However, it turns out that the agent learns to determine the target basket location at a lower height, where only portions of the baskets are in view.



(a) Diving    (b) Grasping    (c) Rising    (d) Gliding    (e) Dropping

Figure 12: Five key states for the conveyance challenge and skills required by these states.

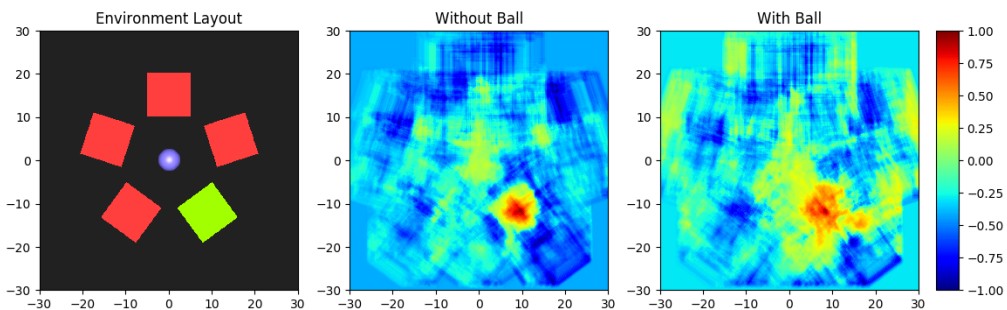

Figure 13: Another tendency heat map of the conveyance task.

E.6   PICK-AND-PLACE DETAILS

(a)

(b)

(c)

(d)

(e)

(f)

Figure 14: Some key states and the corresponding observations in the pick-and-place challenge.

## F   OTHER DISCUSSIONS

The regular reverse curriculum algorithm usually fails if there exists an **irreversible process** in the system. The irreversible process is defined as:

$$\exists s, s^{'} \in \mathcal{S} : \ (\exists n > 0 : \ P(s_n = s'|s_0 = s) > 0) \wedge (\forall n > 0 : \ P(s_n = s|s_0 = s') = 0) \quad (6)$$

In such cases, the states $s$ and $s'$ are not connected, and an agent starting from $s'$ will never reach $s$ since that probability is $0$. We define a **absorbing state** $s_a$ as a state that satisfies

$$P(s_a|s_a) = 1 \wedge \forall s \in \mathcal{S} : \ s \neq s_a \longrightarrow P(a|s) = 0 \quad (7)$$

To be more generalized, we define a set $\mathcal{S}_a$ of states to be a **absorbing set** if it satisfies

$$P(s'|s) = 0 \text{ if } s \in \mathcal{S}_a \wedge s' \notin \mathcal{S}_a \quad (8)$$

Consider a phase extension progress where $\mathcal{P}_{i+1}$ is generated from $\mathcal{P}_i$, if a large portion of states in $\mathcal{P}_{i+1}$ belong to some absorbing sets, it would be hard to for the new phase to include elements not in these absorbing sets. Therefore, the training is likely to be contained within these sets and no actual progress could be made since the phase extension makes no sense.

However, with additional keyframes, this limitation could be avoided even with irreversible processes and absorbing sets. The mechanism is described as follows: When we sample states nearby a keyframe that is not in any absorbing set, the sampled states might happen to belong to some absorbing set. Although phase extensions are constrained within the absorbing sets, the generated phase might also cover some states sampled nearby a keyframe. Thus, according to the phase administrator algorithm (Alg. 2), that keyframe could still be reached.

