# OpenReview forum: "TRL: Discriminative Hints for Scalable Reverse Curriculum Learning"
_ICLR.cc/2018/Conference — Reject_

### Official Review · AnonReviewer1 · 2017-11-27
**Review: Concerns with regard to clarity and real-world applicability**

**Rating:** 4
**Confidence:** 4

**Review:**

This paper proposes a new method for reverse curriculum generation by gradually reseting the environment in phases and classifying states that tend to lead to success. It additionally proposes a mechanism for learning from human-provided "key states".

The ideas in this paper are quite nice, but the paper has significant issues with regard to clarity and applicability to real-world problems:
First, it is unclear is the proposed method requires access only high-dimensional observations (e.g. images) during training or if it additionally requires low-dimensional states (e.g. sufficient information to reset the environment). In most compelling problems settings where a low-dimensional representation that sufficiently explains the current state of the world is available during training, then it is also likely that one can write down a nicely shaped reward function using that state information during training, in which case, it makes sense to use such a reward function. This paper seems to require access to low-dimensional states, and specifically considers the sparse-reward setting, which seems contrived.
Second, the paper states that the assumption "when resetting, the agent can be reset to any state" can be satisfied in problems such as real-world robotic manipulation. This is not correct. If the robot could autonomously reset to any state, then we would have largely solved robotic manipulation. Further, it is not always realistic to assume access to low-dimensional state information during training on a real robotic system (e.g. knowing the poses of all of the objects in the world).
Third, the experiments section lacks crucial information needed to understand the experiments. What is the state, observation, and action space for each problem setting? What is the reward function for each problem setting? What reinforcement learning algorithm is used in combination with the curriculum and tendency rewards? Are the states and actions continuous or discrete? Without this information, it is difficult to judge the merit of the experimental setting.
Fourth, the proposed method seems to lack motivation, making the proposed scheme seem a bit ad hoc. Could each of the components be motivated further through more discussion and/or ablative studies?
Finally, the main text of the paper is substantially longer than the recommended page limit. It should be shortened by making the writing more concise.

Beyond my feedback on clarity and significance, here are further pieces of feedback with regard to the technical content, experiments, and related work:
I'm wondering -- can the reward shaping in Equation 2 be made to satisfy the property of not affecting the final policy? (see Ng et al. '09) If so, such a reward shaping would make the method even more appealing.
How do the experiments in section 5.4 compare to prior methods and ablations? Without such a comparison, it is impossible to judge the performance of the proposed method and the level of difficulty of these tasks. At the very least, the paper should compare the performance of the proposed method to the performance a random policy.

The paper is missing some highly relevant references. First, how does the proposed method compare to hindsight experience replay? [1] Second, learning from keyframes (rather than demonstrations) has been explored in the past [1]. It would be preferable to use the standard terminology of "keyframe".

[1] Andrychowicz et al. Hindsight Experience Replay. 2017
[2] Akgun et al. Keyframe-based Learning from Demonstration. 2012

In summary, I think this paper has a number of promising ideas and experimental results, but given the significant issues in clarity and significance to real world problems, I don't think that the current version of this paper is suitable for publication in ICLR.

More minor feedback on clarity and correctness:
- Abstract: "Deep RL algorithms have proven successful in a vast variety of domains" -- This is an overstatement.
- The introduction should be more clear with regard to the assumptions. In particular, it would be helpful to see discussion of requiring human-provided keyframes. As is, it is unclear what is meant by "checkpoint scheme", which is not commonly used terminology.
- "This kind of spare reward, goal-oriented tasks are considered the most difficult challenges" -- This is also an overstatement. Long-horizon tasks and high-dimensional observations are also very difficult. Also, the sentence is not grammatically correct.
- "That is, environment" -> "That is, the environment"
- In the last paragraph of the intro, it would be helpful to more clearly state what the experiments can accomplish. Can they handle raw pixel inputs?
- "diverse domains" -> "diverse simulated domains"
- "a robotic grasping task" -> "a simulated robotic grasping task"
- There are a number of issues and errors in citations, e.g. missing the year, including the first name, incorrect reference
- Assumption 1: \mathcal{P} has not yet been defined.
- The last two paragraphs of section 3.2 are very difficult to understand without reading the method yet
- "conventional RL solver tend" -> "conventional RL tend", also should mention sparse reward in this sentence.
- Algorithm 1 and Figure 1 are not referenced in the text anywhere, and should be
- The text in Figure 1 and Figure 3 is extremely small
- The text in Figure 3 is extremely small

---

> ### Author Response · Authors · 2017-12-29
> **Rebuttal**
>
> 1-We respectfully remind that full sentence in our paper is “When resetting, the agent can start from any state s ∈ Pi.” We don’t assume that the agent can reset to any state. Actually, we only assume that it can reset to a certain state in each phase where it has reached before. Thanks for mentioning the access to low-dimensional states. TRL does need these low-dimensional data to restore visited states during the generation of new phases and doesn’t require these data for real training. During each generation process, the newly sampled states will be stored in the form of low-dimensional states such as the angle of joints and velocity of motors. Since these low-dimensional data is easy to acquire and only used for resetting the agent, we just summarized it as “a way of adding new states to the new phase”. It seems that there is no need for special emphasize.
>
> 2-As is mentioned in the last paragraph of Introduction: “The major contribution of this work is that we present a reliable tendency reinforcement learning method that is capable of training agents to solve large state space tasks with only final reward. ” This is our reward setting and is just the definition of goal-oriented tasks. And the detail of experiments is also shown in Appendix C, where we explain all of the settings. The RL used in all of our experiments is A3C and our action control is discrete.
>
> 3-There are three components: (a) Phase administrator (b) Tendency reward (c) Keyframes ("checkpoint" is renamed "keyframe")
> We ran rough ablation studies with three different settings of difficulties:
> (i) small state space with only final reward (10*10 Maze with observation 10*10): None of the three components are needed since a traditional RL method can tackle it.
> (ii) medium state space with only final reward (40*40 Maze with observation 9*9, Mario Bros): We can solve it by only using (b) with around 53000 training steps(40*40 Maze). We can also accelerate learning by combine (b) and (a), which will take around 35000 training steps.
> (iii) large state space with only final reward (100*100 Maze with observation 9*9, robotic manipulation from perception(grasping, pick and place)): We use (a), (b)and (c) to solve these problems. If we only use (a) and (b), the generation of each phase might be biased and will fail in multistage tasks. Then we include (c) and test the influence of keyframes with different quality and scale (Appendix E.3 E.4 Fig 10 11). We do not find clear relationship between the number of keyframes and the efficiency of training, but keyframes can indeed help TRL learn well (33000 iterations in Grasping, 99000 iterations in Conveyance challenge).
>
> 4-We ran some tests based on [Ng et al 1999] and found that if we structure the tendency reward as potential based, the efficiency will largely decrease. We tested it in 40*40 Maze with observation 9*9. Since the tendency reward then be defined as rT(St’) - T(St), the hybrid reward is still very sparse and the agent takes more than 50000 iterations to complete 60% of the whole task (our method takes around 35000 steps to complete the whole one).
>
> 5-Goal-oriented tasks are among the most difficult challenges in RL and traditional methods (e.g. TRPO, AC, PPO) alone are not capable of tackling them. The most recent approach to tackle it is based on intrinsic motivation. We made an experiment comparing TRL with curiosity-driven RL in Appendix E.1.2 (Table 2) and showed TRL’s advantages. Other methods mainly focus on tackling this problem with demonstrations, which we also compare TRL with in Experiment 5.3 (Fig 6). The result shows that we only need a small number of keyframes to achieve better results compared to them without much human elaboration or well hand-engineered reward function.
>
> 6-Thanks. We have incorporated these two works in discussion.

---

> > ### Comment · AnonReviewer1 · 2018-01-10
> > **Response to rebuttal**
> >
> > > 1a. We respectfully remind that full sentence in our paper is “When resetting, the agent can start from any state s ∈ Pi.” We don’t assume that the agent can reset to any state
> >
> > Thank you for the clarification. However, assuming resets even in Pi is not practical in many robotic manipulation problems, e.g. any problem involving free moving objects such as pushing or pick and place (e.g. when the robot must learn to also move the object back to where it started).
> >
> >
> > > 1b. TRL does need these low-dimensional data to restore visited states during the generation of new phases and doesn’t require these data for real training… Since these low-dimensional data is easy to acquire…
> >
> > I agree that joint angle and end-effector information is easy to acquire. But in practice, *full* low-dimensional state information is not easy to acquire (i.e. positions of free moving objects) and if you assume access to it during some parts of training, then you might as well use it for all parts of training. For example, imagine you wanted to apply this method to a robot learning pushing an object (a fairly simple task). You would need to put some sort of tracker on the object to get its low-dimensional state. If you need to put a tracker on the object, then you might as well use the tracker during training too.
> >
> >
> > Thank you for running the additional experiments. I think that they improve the paper.

---

### Official Review · AnonReviewer3 · 2017-11-27
**Interesting idea, but approach seems limited.**

**Rating:** 4
**Confidence:** 4

**Review:**

The authors extend the approach proposed in the  "Reverse Curriculum Learning for Reinforcement Learning" paper by adding a discriminator that gives a bonus reward to a state based on how likely it thinks the current policy is to reach the goal from said state. The discriminator is a potentially interesting mechanism to approximate multi-step backups in sparse-reward environments.

The approach of this paper seems severely severely limited by the assumptions made by the authors, mainly assuming a deterministic environment, known goal states and the ability to sample anywhere in the state space. Some of these assumptions may be reasonable in domains such as robotics, but they seem very restrictive in the domains like the games considered in the paper.


Additional Comments:

-The authors demonstrate some benefits of using Tendency rewards, but made little attempt to explain why it leads to accelerated learning. Results are pure performance results.

-The authors should probably structure the tendency reward as potential based instead of using the Gaussian kernel hack they introduce in section 4.2

- Presentation: There are several mistakes and formatting issues in References

- Assumption 2 transformations -> transitions?

-Need to add assumption 3: advance knowledge of goal state

- the use of gamma as  a scale factor in equation 2 is confusion, it was already introduced as the discount factor ( which is default notation in RL). It also isn't clear what the notation r_f denotes (is it the same as r^f in appendix?).

-It is nice to see that the authors compare their method with alternative approaches. Unfortunately, the proposed method does not seem to offer many benefits.

---

> ### Author Response · Authors · 2017-12-29
> **Rebuttal**
>
> 1-As is claimed in the paper, our assumption follows [Carlos et al 2017]. For deterministic environments, we found it not necessary since we can change the discriminator to the probability of success between 0-1 and TRL can then handle stochastic as well. We have revised the claim. For the sample-anywhere assumption, in fact, we don’t need to reach everywhere but only start states in the current phase which it has reached during the generation process. We can record those states through low dimensional data (angles of joint etc) easily. In games, actually we find it’s easier than robotics to reset to any state given access to the corresponding API from developers. Given that many game developers are interested in training AI agent automatically for their games, such APIs are usually not hard to acquire.
>
> 2-As is explained in the Introduction, we have pointed out the reason why the method in Reverse Curriculum paper is lack of efficiency (Close to the end of the 2nd paragraph). Then we show that with the help of tendency rewards, our model can get rid of the unnecessary time-consuming reviewing process where the agent switches start states between old and new ones to avoid forgetting old policies (End of the 3rd & middle of the 4th paragraph). To prove our idea, we make a comparison in Experiment 5.1 (Fig 3), which shows our advantage in efficiency compared to Reverse Curriculum algorithm. TRL’s main advantage over reverse curriculum is that it no longer requires keeping all starting sets.
>
> 3-Thanks for mentioning the potential based reward shaping. However, if we define the shaped reward as r = T(St’) - T(St), although this approach can avoid repeated rewards, it still suffer from the reward sparsity problem, since T(St’_positive) - T(St_positive) and T(St’_negative)- T(St_negative) remain 0 at most time and won’t help the agent learn to tackle these tasks.
>
> 4-Thanks! We have added this assumption. This assumption is also listed in [Carlos et al 2017].
>
> 5-This gamma is only used for weight balance for two rewards. We are sorry to use a confusing notation. Another notation $\lambda$ has been used to address the confusion.
>
> 6-We explained in Experiment 5.3 that the reward function used in PBRS is well hand-engineered by us. We tried more than 10 different reward functions shaped from demonstration and keep adjusting them to let PBRS solve this task. In our experiments, only 2 of all the reward functions we tried can let PBRS work, the others are not shown on the Fig 6. This approach costs much human elaboration and different maps in the Maze need different reward function. Moreover, in most robotic domains where the reward function cannot be easily shaped by hands, human elaboration will increase to an unpractical level. TRL is able to solve this problem with negligible human elaboration with merely several labeled keyframes ("checkpoint" is renamed "keyframe"). We also proved TRL’s robustness to keyframes with different quality and scale in Appendix E.3 & E.4 (Fig 10, Fig 11). Although the training efficiency of TRL and PBRS may seem similar in the figure, the human elaboration behind the performance is quite different.

---

### Official Review · AnonReviewer2 · 2017-12-01
**interesting direction, questions about the proposed approach**

**Rating:** 5
**Confidence:** 4

**Review:**

The authors present a new method for doing reverse curriculum training for reinforcement learning tasks with deterministic dynamics, a desired goal state at which reward is received, and the ability to teleport to any state. This covers a number of important cases of interest, including all simulated domains, and a number of robotics applications. The training proceeds in phases, where in each phase the initial starting set of states is expanded. The initial set of states used is close to the desired state goal. Each phase is initiated when 80% of the states in the current phase can reach the goal. Once the initial set of start states overlaps with the desired initial set of states for the task, training can terminate. During the training in a single phase, the algorithm uses a shaping reward (the tendency) which is based on a binary classifier that predicts if it will be possible to reach the goal from this state. This reward is combined in a hybrid reward signal. The authors suggest the use of a small number of checkpoints to guide the backwards state expansion to improve the search efficiency. Results are presented on several domains: maze, Super Mario, and Mujoco domains.

The topic of doing more sample efficient training is important and interesting, and the subset of settings the authors consider is still a good set.

The paper was clearly written though some details were relegated to the appendix which would’ve been useful to see in the main text.

I’m not yet convinced about this method for the desired setting in terms of significance and quality.

An alternative to using tendency shaping reward would be (during phase expansion) make the new “goal” states any of the states in the previous phase of initial states P_{i} that did reach the goal. This should greatly reduce the decision making horizon needed in each  phase. Since the domain is deterministic, as soon as one can reach one of those states, we have a path to the goal. If we care about the number of steps to reach the goal (vs finding any path), then each of the states in P_{i} for which a successful path can be achieved to the goal can also be labeled by the cost / number of time steps to reach the goal. This should decompose the problem into a series of smaller problems. Perhaps I’m missing something-- could the authors please address this suggestion and/or explain why this wouldn’t be beneficial?

The authors currently use checkpoints to help guide the search towards the true task desired set of initial states. If those are lacking, it seems like the generation of the new P_{i+1} could be biased towards that desired set of states. One approach could be to randomly roll out from the start state and then bias P_{i+1} towards any states close to states along such trajectories. In general one could imagine a situation in which one both does forward learning/planning from the task start state and backwards learning from the goal state to obtain a significant benefit, similar to ideas that have been used in robot motion planning.

Why learn from pixels for the robot domains considered? Here it would be nice to compare to some robotics approaches. With the action space of the robot and motion planning, it seems like this problem could be tackled using existing techniques. It is interesting to have a method that can be used with pixels, but in cases where there are other approaches, it would be useful to compare to them.

Small point
D.2 Why not compare to GAIL instead?

---

> ### Author Response · Authors · 2017-12-29
> **Response to AnonReviewer2**
>
> 1-Thanks for mentioning that. Actually, this alternative has been carefully considered, and we decided not to use it mainly because this method largely impairs the agent's ability to find new policies. We tested this idea with an experiment setup similar to the one in Appendix E.3 (Fig 10), and found that if we change the goal state to any of the successful states from the previous phase, the agent is highly likely to lose the capability of finding a new shortcut (the fifth graph in Fig 10). The reason is that TRL's reward function is hybrid (tendency + final goal), where the final goal reward is meant to guarantee the agent's motivation for finding new policies. That’s why keeping the final goal state constant in training of each phase makes sense.
>
> 2-Thanks for your suggestion. Based on some experiments on this idea, we find that in small state space tasks (e.g. the Maze) this approach can lead to similar performance compared to keyframe scheme ("checkpoint" is renamed "keyframe"), but it might be impractical in large state space multistage tasks such as “Pick and Place”. Since the shaping of tendency reward hasn’t covered the area close to the start state, exploration beginning from the start state might be biased as well, and the complexity of generating P_{i+1} can be very high. As a matter of fact, several keyframes can already solve this problem well in these domains.
>
> 3-We learn from raw pixel perceptions based on the assessment that it is a more general form of environment information and contains more details of the environment than low-dimensional data. Classic approaches, due to hand-designed detectors and grasp policies, cannot be easily generalized to new objects or varying background scenes. Additionally, images are less expensive to acquire and are more practical than precise sensor information. Taking robotic grasping and picking as an example, the location and shape of the object are hard to acquire and define (we cannot mount sensors everywhere), we will have to rely on perceptions (image or video).
>
> 4-TRL does NOT fall in the track of imitation learning. The optional keyframes are only used in large-scale experiments like grasping from perception, not in simpler ones like Mario and Maze. By our design, TRL works without any expert policies. The keyframe scheme only helps to shrink search space and does not influence the learned policy. Our experiments show that the agent does not necessarily follow the keyframes (Appendix E.3 Fig 10).

---

### Decision · Program_Chairs · 2018-01-29
**ICLR 2018 Conference Acceptance Decision**

**Decision:**

Reject

**Comment:**

The paper proposes an extension to the reverse curriculum RL approach which uses a discriminator to label states as being on a goal trajectory or off the goal trajectory. The paper is well-written, with good empirical results on a number of task domains. However, the method relies on a number of assumptions on the ability of the agent to reset itself and the environment which are unrealistic and limiting, and beg the question as to why use the given method at all if this capability is assumed to exist. Overall, the method lacks significance and quality, and the motivation is not clear enough.